# Is Sortition Both Representative and Fair?

**Soroush Ebadian**
Department of Computer Science
University of Toronto
soroush@cs.toronto.edu

**Gregory Kehne**
School of Engineering and Applied Sciences
Harvard University
gkehne@g.harvard.edu

**Evi Micha**
Department of Computer Science
University of Toronto
emicha@cs.toronto.edu

**Ariel D. Procaccia**
School of Engineering and Applied Sciences
Harvard University
arielpro@seas.harvard.edu

**Nisarg Shah**
Department of Computer Science
University of Toronto
nisarg@cs.toronto.edu

## Abstract

*Sortition* is a form of democracy built on random selection of representatives. Two of the key arguments in favor of sortition are that it provides *representation* (a random panel reflects the composition of the population) and *fairness* (everyone has a chance to participate). Uniformly random selection is perfectly fair, but is it representative? Towards answering this question, we introduce the notion of a representation metric on the space of individuals, and assume that the cost of an individual for a panel is determined by the $q$-th closest representative; the representation of a (random) panel is measured by the ratio between the (expected) sum of costs of the optimal panel for the individuals and that of the given panel. For $k/2 < q \leq k - \Omega(k)$, where $k$ is the panel size, we show that uniform random selection is indeed representative by establishing a constant lower bound on this ratio. By contrast, for $q \leq k/2$, no random selection algorithm that is almost fair can give such a guarantee. We therefore consider relaxed fairness guarantees and develop a new random selection algorithm that sheds light on the tradeoff between representation and fairness.

## 1 Introduction

Most people think of democracy as synonymous with elections. But that has not always been the case: from the inception of democracy in ancient Athens until the American and French revolutions, democracy had typically been associated with random selection of representatives [1], a paradigm known as *sortition*.

These days, sortition is mainly practiced in the form of *citizens' assemblies* — randomly selected groups of people who deliberate on central questions, with the goal of generating recommendations and informing policy. The impact and prevalence of citizens' assemblies around the world have motivated computational work on how to fairly and transparently select assembly members [2–4]. But there are signs that sortition is becoming even more widely accepted, including its recent institutionalization in Belgium, where permanent sortition-based bodies are now working alongside the parliaments of the German-speaking region and the Brussels region. In light of this progress, it

36th Conference on Neural Information Processing Systems (NeurIPS 2022).

may only be a matter of time until one of the many blueprints for sortition-based democracy [5] is implemented at the level of an entire country.

The excitement about sortition is driven by several appealing qualities, which are seen as providing solutions to some of the problems plaguing electoral democracy. We briefly present two of them in the context of *uniform selection*, which selects a uniformly random panel and is considered to be the ideal sortition method [6].

- *Descriptive representation*: A panel selected uniformly at random is likely to reflect the composition of the population from which it was drawn. Representation lends *legitimacy* to the process [7, 8], as individuals are able to identify some panelists who are similar to themselves.[1]
- *Fairness*: Under uniform selection, each individual has an equal chance to participate. Political theorists have argued that this quality realizes philosophical ideals like equality of opportunity and allocative justice [10].

By any reasonable measure of the fairness of selection probabilities — e.g., the minimum selection probability of any individual [2] — uniform selection achieves perfect fairness, as selection probabilities are equalized. We ask: *Is uniform selection also representative in a rigorous sense?* If we had an analogous measure of representation, we would be able to explore whether this is the case. But quantifying representation poses a conceptual challenge.

**Our approach.** We address this challenge by assuming that there exists a *representation metric* on individuals, which measures to what degree one individual represents another (smaller distance means better representation). Readers familiar with the algorithmic fairness literature will no doubt make the connection to the *similarity metric* of Dwork et al. [11], which has been criticized on the grounds that it is difficult to explicitly construct [12]; a major obstacle is that the question of whether certain features should be used to determine similarity is domain-specific and tied to legal interpretation. By contrast, the prospect of constructing a representation metric is perhaps more tractable for practitioners. This is because the organizers of a citizens assembly (say) can choose it to be a function of a common set of features which are already routinely used by practitioners for the purpose of guaranteeing the representation of selection rules. Moreover, some of our main results, which pertain to uniform selection, are fully independent of the metric — for these results it suffices to suppose that such a metric *exists*.

Note, however, that a distance metric (on individuals) does not directly define a measure of the degree to which an individual is represented by a panel. Following very recent work by Caragiannis et al. [13], we assume that the *cost* of a panel for an individual is determined by the $q$-th closest member of the panel, and our results are parameterized by $q$.

We can now define representation by taking a page from the literature on *distortion* in social choice [14]. Specifically, for a given selection algorithm, we measure its representation via the ratio between the social cost (sum of costs) of the optimal panel and that of the panel chosen by the algorithm, *in the worst case over underlying representation metrics*.

**Our results.** Returning to the question of whether uniform selection is also representative, and, more generally, the eponymous question of whether sortition is both representative and fair, our answer is that "it depends" — on the value of $q$.

When $k/2 < q \leq k - \Omega(k)$, where $k$ is the size of the panel, we show that uniform selection (which is perfectly fair) achieves constant representation. Qualitatively, we view this as providing positive answers to our questions in the regime of $q > k/2$. Note that this regime has a natural interpretation: each individual wants a majority of the panel to be representative of themselves. This is especially justifiable when the panel makes decisions or recommendations through voting, which is often the case in citizens' assemblies.

By contrast, for the regime of $q \leq k/2$, we prove that any selection algorithm that chooses each individual with probability somewhat higher than $q/n$, where $n$ is the number of individuals, must in the worst case have representation of precisely 0. This result clearly applies to uniform selection, where

---

[1]Another motive for representation is epistemic: [9] and others argue that a diversity of opinions leads to better decisions.

the minimum selection probability is $k/n$, and it motivates us to consider weaker fairness guarantees. We design an algorithm, RANDOMREPLACE, which selects each individual with probability at least $q/n$ and has a nontrivial representation guarantee of $1/(q+1)$ for any value of $q$.

Finally, we evaluate the average-case representation of uniform selection and RANDOMREPLACE on inferred metrics derived from two demographic datasets. For $q > k/2$, our experiments show that uniform selection achieves representation consistently greater than 55%. Even for $q \leq k/2$, in contrast to our worst-case result, uniform selection achieves good average-case representation. These results suggest that, in practice, uniform selection may be a good choice for all $q$-cost distance functions. While RANDOMREPLACE outperforms uniform selection across the board in terms of representation (at the expense of fairness), its advantage is small for values of $q$ around $k/2$ and close to $k$, which means it does not offer a good representation-fairness tradeoff in those regimes. Lastly, for both algorithms, we observe a spike in representation at $q = k/2$, which demonstrates that the chasm between provable guarantees for $q \leq k/2$ versus $q > k/2$ is not merely a theoretical curiosity.

**Related work.** The design of practical, fair and transparent algorithms for selecting citizens' assemblies was explored in several previous papers [2–4]—two of which appeared in previous NeurIPS conferences. Assemblies are required to be representative of the population with respect to features like gender, age, ethnicity, education and geography. This is generally done by setting quotas on individual features; for example, a panel of 100 people might be required to include at least 48 men, at least 48 women, and at least 2 people who identify as non-binary. The challenge is that an assembly is selected from a pool of volunteers, and this pool is typically unrepresentative of the population due to self-selection bias. Uniformly random selection from the pool, therefore, would likely itself result in an unrepresentative panel. Instead, the primary selection algorithm advocated by Flanigan et al. [2] computes a distribution over quota-compliant panels that (roughly speaking) maximizes the minimum selection probability of any volunteer, thereby maximizing fairness subject to these hard demographic constraints. By contrast, like Benadè et al. [15] and the political theory literature, we take a longer-term view: We are interested in random selection directly from the population, which is a hallmark of some plans for sortition-based democracy [5]. In addition, we take a fundamentally different, and arguably more nuanced, view of representation. Indeed, our framework can theoretically accommodate considerations of intersectionality (to what degree is a rural, college-educated man represented by a rural, college-educated woman?) and could also capture a more holistic analysis of the composition of the panel (to what degree is a rural, college-educated man represented by a panel that includes 10 rural, college-educated men and 90 urban women with no college education?). Lang and Skowron [16], Celis et al. [17], and Do et al. [18] also study the problem of creating committees or assemblies in which the features of the participants satisfy some desired quotas, but in the absence of an underlying representation metric and any fairness constraints.

Our approach to evaluating representation through a metric is rooted in spatial theories of voting from political theory [19, 20]. The idea of measuring how poorly a panel represents an individual by the distance of the $q$-th closest panel member to the individual was introduced by Caragiannis et al. [13] in the context of committee elections. Finally, aiming to minimize the total misrepresentation across all people and comparing that to the optimal panel makes our representation measure the inverse of *distortion* in voting theory [21, 22]. In voting, it is assumed that we only have partial access to the metric in the form of voters' ranked preferences over the candidates, which are induced by distance comparisons. In contrast, our results for uniform selection use no knowledge of the underlying metric, while our other algorithms assume complete access. When selecting a single candidate as the winner, it is known that the best distortion achievable by deterministic selection is 3 (which maps to $1/3$ representation in our formulation) [23]. Our setting is closer to committee selection, where a committee of $k$ candidates is selected. Here, Caragiannis et al. [13] show that when each voter measures her distance to the $q$-th closest committee member, there is a trichotomy: the best possible distortion is infinite when $q \leq k/3$, linear in the number of voters when $q \in (k/3, k/2]$, and 3 for deterministic selection when $q > k/2$. Sortition may be viewed as a special case of randomized committee elections, in which the set of candidates is the same as the set of voters. Hence, all the positive results from Caragiannis et al. [13] carry over in the absence of any fairness constraints. However, our results show that when (perfect) fairness in selection probabilities is sought in conjunction with representation, the distortion becomes infinite (zero representation) for all $q \leq k/2$ but constant distortion can still be achieved for $q > k/2$. The idea of the set of voters acting as the set of candidates was explored by Cheng et al. [24, 25]. However, they model infinitely many voters using a continuous distribution over the metric space.

## 2 Preliminaries

For all $t \in \mathbb{N}$, define $[t] = \{1, \ldots, t\}$. Let $N = [n]$ be the set that indexes the underlying population. A *panel* $P$ is a subset of the population. Let $\mathcal{S}_k(N)$ denote the set of all subsets of $N$ of size $k$. (We omit $N$ when it is clear form the context.) The population lies in a metric space with distance $d$, which we view as the *representation metric* discussed earlier. For each $i, j \in N$, $d(i, j)$ denotes the distance between $i$ and $j$; $d$ is a metric if the following properties are satisfied: (a) $d(i, j) \geq 0$, and $d(i, j) = 0$ if and only if $i = j$, (b) $d(i, j) = d(j, i)$, and (c) for each $i, j, \ell \in N$, $d(i, \ell) + d(\ell, j) \geq d(i, j)$. The last property is known as the triangle inequality. An instance of our problem is given by the underlying population along with distances as defined by $d$; hereinafter, we simply denote such an instance by $d$.

Given a panel $P$ of size $k$ and a positive integer $q \in [k]$, the *q-cost* of individual $i$ for $P$, denoted by $c_q(i, P; d)$, is equal to the distance of $i$ from her $q$-th closest representative in $P$. Note that for $q = 1$, we have $c_1(i, P; d) = \min_{j \in P} d(i, j)$ and for $q = k$ we get $c_k(i, P; d) = \max_{j \in P} d(i, j)$. Let $\mathsf{top}_q(i, P; d)$ be the set of $q$ closest members of $P$ to $i$ (with ties broken arbitrarily). The *q-social cost* of panel $P$ is given by $\mathrm{SC}_q(P; d) = \sum_{i \in N} c_q(i, P; d)$, i.e., the sum of the $q$-costs over all individuals. (Observe that if $q = 1$ then $\mathrm{SC}_q(P; d)$ is the standard $k$-medians clustering objective evaluated for centers $P$.) We omit $d$ from the notation when it is clear from the context.

In this setting, a *selection algorithm* $\mathcal{A}_{k,q}$ parameterized by $k$ and $q$ takes as input the metric $d$ and outputs a distribution over all panels of size $k$. We are especially interested in the *uniform selection* algorithm, denoted by $\mathcal{U}_k$, that always outputs a uniform distribution over $\mathcal{S}_k$, regardless of $q$. In other words, it does not take into account the underlying metric space or $q$, but instead outputs a committee of size $k$ chosen uniformly at random. We now formally introduce the fairness and representation of a selection algorithm.

**Fairness:** One appealing property of uniform selection is that each individual is selected to be part of the panel with probability exactly equal to $k/n$, i.e., $\Pr[i \in \mathcal{U}_k] = \frac{k}{n}$. In particular, all individuals have an equal chance of being chosen. We call this property *perfect fairness*; in general an algorithm $\mathcal{A}_{k,q}$ provides perfect fairness when for each instance $d$, it ensures that $\min_{i \in N} \Pr[i \in \mathcal{A}_{k,q}] = k/n$.

When perfect fairness is too restrictive, we relax this constraint by allowing individuals to be selected with probability less than $k/n$. In this case, the fairness of an algorithm $\mathcal{A}_{k,q}$ is the worst-case ratio of the minimum probability of an individual to be selected by the algorithm and the ideal selection probability of $k/n$. Formally,

$$\mathsf{fairness}_q(\mathcal{A}_{k,q}) = \inf_d \frac{\min_{i \in N} \Pr[i \in \mathcal{A}_{k,q}(d)]}{k/n}.$$

**Representation:** The other key property we would like to measure is representation. To do so, we consider a panel to be a good representative of the whole population when its $q$-social cost is not much larger than the best possible. In other words, a selection algorithm $\mathcal{A}_{k,q}$ that outputs a distribution over the different committees of size $k$ provides good representation when the expected $q$-social cost of the panel is similarly small. More formally, we define the *representation* of a selection algorithm $\mathcal{A}_{k,q}$ as the worst-case ratio of the minimum possible social $q$-cost of any panel and the expected $q$-social cost of the panel chosen by $\mathcal{A}_{k,q}$ over all possible instances; i.e.,

$$\mathsf{repr}_q(\mathcal{A}_{k,q}) = \inf_d \frac{\min_{P' \in \mathcal{S}_k(N)} \mathrm{SC}_q(P'; d)}{\mathbb{E}[\mathrm{SC}_q(\mathcal{A}_{k,q}(d)]}.$$

## 3 Representation with Perfect Fairness for $q > k/2$

We begin by considering the case that $q > k/2$. We show that in this case uniform selection is asymptotically optimal with respect to representation among all selection algorithms that are perfectly fair. Moreover, the representation of uniform selection is constant for any $q = c \cdot k$ for $1/2 < c < 1$.

**Theorem 1.** *For $q > k/2$, uniform selection satisfies* $\mathsf{repr}_q(\mathcal{U}_k) \geq \frac{1}{2} \cdot \frac{k-q+1}{k}$.

A crucial property that we exploit in this section is that the $q$-costs of the individuals satisfy the triangle inequality when $q > k/2$. This observation was first made by Caragiannis et al. [13]; we present the lemma below for completeness.

**Lemma 1.** *For $q > k/2$, individuals $i, j \in N$, and a panel $P$, $c_q(i, P; d) + c_q(j, P; d) \geq d(i, j)$.*

*Proof.* Let $T_i = \text{top}_q(i, P)$ and $T_j = \text{top}_q(i, P)$ be the $q$ closest neighbors of $i$ and $j$, respectively, in the panel $P$. As $|T_i| = |T_j| > k/2$, there exists an individual $k \in T_i \cap T_j$. Therefore,

$$d(i, j) \leq d(i, k) + d(k, j) \leq c_q(i, P) + c_q(j, P). \qquad \square$$

We use this observation to lower bound the social cost of the optimal committee.

**Lemma 2.** *For $q > k/2$, the $q$-social cost of the optimal panel $P^*$ is at least*

$$\text{SC}_q(P^*; d) \geq \frac{1}{2(n-1)} \cdot \sum_{i \in N} \sum_{j \in N \setminus \{i\}} d(i, j).$$

*Proof.* By applying Lemma 1 for all pairs of individuals $(i, j)$, we get

$$\sum_{i \in N} \sum_{j \in N \setminus \{i\}} \Big( c_q(i, P^*) + c_q(j, P^*) \Big) \geq \sum_{i \in N} \sum_{j \in N \setminus \{i\}} d(i, j).$$

The $q$-cost of each person appears exactly $2(n-1)$ times on the left hand side. Thus,

$$\text{SC}_q(P^*; d) = \sum_{i \in N} c_q(i, P^*) \geq \frac{1}{2(n-1)} \sum_{i \in N} \sum_{j \in N \setminus \{i\}} d(i, j). \qquad \square$$

We are now ready to prove the theorem.

*Proof of Theorem 1.* For any committee $P$ of size $k$,

$$c_q(i, P) = \min_{q' \in \{q, \dots, k\}} c_{q'}(i, P) \leq \frac{1}{k - q + 1} \sum_{q' \in [q, k]} c_{q'}(i, P)$$

$$\leq \frac{1}{k - q + 1} \sum_{q' \in [1, k]} c_{q'}(i, P) = \frac{1}{k - q + 1} \sum_{j \in P} d(i, j),$$

where in the first inequality the minimum is upper bounded by the average. Therefore, the expected social cost of uniform selection is at most

$$\mathbb{E}[\text{SC}_q(\mathcal{U}_k(N))] = \sum_{i \in N} \mathbb{E}_{P \sim \mathcal{U}_k} [c_q(i, P)]$$

$$\leq \frac{1}{k - q + 1} \sum_{i \in N} \mathbb{E}_{P \sim \mathcal{U}_k} \Big[ \sum_{j \in P} d(i, j) \Big]$$

$$= \frac{1}{k - q + 1} \sum_{i \in N} \sum_{j \in N \setminus \{i\}} d(i, j) \cdot \text{Pr}_{P \sim \mathcal{U}_k} [j \in P]$$

$$= \frac{1}{k - q + 1} \sum_{i \in N} \sum_{j \in N \setminus \{i\}} d(i, j) \cdot \frac{k}{n}.$$

By Lemma 2 and the upper bound shown above, we have

$$\text{repr}_q(\mathcal{U}_k) \geq \frac{\frac{1}{2(n-1)} \sum_{i \in N} \sum_{j \in N \setminus \{i\}} d(i, j)}{\frac{1}{k-q+1} \cdot \frac{k}{n} \cdot \sum_{i \in N} \sum_{j \in N \setminus \{i\}} d(i, j)} = \frac{1}{2} \cdot \frac{n}{n-1} \cdot \frac{k - q + 1}{k} \geq \frac{1}{2} \cdot \frac{k - q + 1}{k}. \quad \square$$

In the proof above, the only property of uniform selection we use is that the marginal probabilities are equal to $k/n$. Hence, this lower bound also holds for any perfectly fair selection algorithm.

We next establish an upper bound on the representation of any perfectly fair selection algorithm. It shows that the lower bound of Theorem 1 is tight up to a factor of 4.

**Theorem 2.** *For any $q > k/2$, every selection algorithm $\mathcal{A}_{k,q}$ with $\text{fairness}(\mathcal{A}_{k,q}) = 1$ satisfies $\text{repr}_q(\mathcal{A}_{k,q}) \leq 2 \cdot \frac{k-q+1}{k+1}$.*

*Proof.* First, note that if $q = \frac{k+1}{2}$, the statement trivially holds, since $\text{repr}_q(\mathcal{A}_{k,q}) \leq 2 \cdot \frac{k-q+1}{k+1} = 1$ which is true for any algorithm. Thus, we assume $q > \frac{k+1}{2}$. Consider an instance with $n = k + 1$ individuals where $k - q + 1$ individuals are located at 0 and $q$ individuals are at 1, denoted by $N_0$ and $N_1$, respectively. Any committee of size $k$ leaves one person out of the committee, and for each individual $i \in N$, this happens with probability of

$$\Pr_{P \sim \mathcal{A}_{k,q}}[i \notin P] = 1 - \Pr_{P \sim \mathcal{A}_{k,q}}[i \in P] = (1 - \tfrac{k}{k+1}) = \tfrac{1}{k+1}.$$

Individuals in $N_0$ will always have a $q$-cost of 1, because $k + 1 - q < \frac{k+1}{2} < q$ individuals are located there. Therefore, $\mathbb{E}_{P \sim \mathcal{A}_{k,q}}[\sum_{i \in N_0} c_q(i, P)] = |N_0|$. For individuals in $N_1$, their $q$-cost is 1 if and only if less than $q$ individuals are selected from $N_1$, i.e., the single person left out of the committee is located at 1. This event happens with probability

$$\Pr_{P \sim \mathcal{A}_{k,q}}[\bigcup_{i \in N_1}(i \notin P)] = \sum_{i \in N_1} \Pr_{P \sim \mathcal{A}_{k,q}}[i \notin P] = \tfrac{|N_1|}{k+1},$$

where the first equality comes from the fact that the events are disjoint (which holds because any committee leaves out exactly one individual). Therefore, $\mathbb{E}_{P \sim \mathcal{A}_{k,q}}[\sum_{i \in N_1} c_q(i, P)] = |N_1| \cdot \frac{|N_1|}{k+1}$, and the expected social cost of any perfectly fair algorithm is

$$\mathbb{E}_{P \sim \mathcal{A}_{k,q}}[\text{SC}_q(P; d)] = |N_0| + |N_1| \cdot \frac{q}{k+1} \geq |N_0| + |N_1| \cdot \frac{1}{2}.$$

The optimal committee would leave out a person from $N_0$ and achieve a social cost of $|N_0|$. Therefore, the representation of any algorithm with perfect fairness is at most

$$\frac{|N_0|}{|N_0| + \frac{1}{2} \cdot |N_1|} \leq \frac{|N_0|}{\frac{1}{2} \cdot |N_0| + \frac{1}{2} \cdot |N_1|} = 2 \cdot \frac{k - q + 1}{k + 1}. \qquad \square$$

## 4   Representation with Relaxed Fairness for $q \leq k/2$

In stark contrast to the case of $q > k/2$, uniform selection and, more generally, any perfectly fair selection algorithm, cannot obtain bounded representation when $q \leq k/2$. In fact, the following theorem shows that selection algorithms with fairness strictly more than $\frac{q+(k \bmod q)}{k}$ (which itself is upper bounded by $(2q - 1)/k$) suffer from unbounded representation. The proof is in Appendix A.

**Theorem 3.** *For $q \leq k/2$, $\epsilon > 0$, and any selection algorithm $\mathcal{A}_{k,q}$ with* $\text{fairness}(\mathcal{A}_{k,q}) \geq \frac{q+(k \bmod q)}{k} + \epsilon$, $\text{repr}_q(\mathcal{A}_{k,q})$ *is 0.*

Although bounded representation is not feasible with fairness slightly larger than $\frac{q}{k}$, we design a selection algorithm that, given an $\alpha$-representative panel, can achieve representation of at least $\frac{\alpha}{q+1}$ with fairness of $\frac{q}{k}$. In particular, starting from the optimal panel ($\alpha = 1$), it achieves $\frac{1}{q+1}$ representation. Although finding the optimal panel given the metric space is an NP-hard problem for most metric spaces, computing a constant-factor approximation (i.e., constant $\alpha$) is feasible in polynomial time [26], as we explain in Section 5.

Our Algorithm, RANDOMREPLACE$_q$, is given in Algorithm 1. It starts from a panel $P$, randomly selects a panel $S$ of $q$ individuals, and replaces individuals in $P$ with individuals in $S$ as follows. First, individuals in $S \cap P$ remain in the final panel. Then, for $i \in S \setminus P$, swap $i$ with one of its $q$-closest neighbors in the optimal panel $\text{top}_q(i, P)$ that has not been replaced by the algorithm yet. The next theorem establishes the fairness and representation guarantees of RANDOMREPLACE.

**Theorem 4.** *For any $q \in [k]$ and any panel $P$ with* $\text{repr}_q(P) = \alpha$, *we have that* $\text{repr}_q(\text{RANDOMREPLACE}_q(P)) \geq \frac{\alpha}{q+1}$ *and* $\text{fairness}(\text{RANDOMREPLACE}_q) \geq \frac{q}{k}$.

*Proof.* Let $S \subseteq N$ be a set of size $q$ chosen uniformly at random. We denote with $P_S$ the panel that is returned from the algorithm. First, we show that Line 4 of the algorithm is valid. The algorithm reaches this line when it considers $i \in S \setminus P_S$, meaning that $i$ is not included in the panel $P_S$ and therefore is not included in $P$. Hence, $\text{top}_q(i, P) \setminus S$ cannot be empty since $|\text{top}_q(i, P)| = q$, $|S| = q$ and there exists $i$ in $S$ but not in $\text{top}_q(i, P)$.

---

**ALGORITHM 1:** RANDOMREPLACE$_q$

---

**Input:** Panel $P$ with repr$_q(P) = \alpha$
**Output:** Randomly selected panel by replacing at most $q$ individuals of $P$
1: Pick $S \in \mathcal{S}_q$ uniformly at random
2: Set $P_S \leftarrow P$ and $\overline{S} \leftarrow S \setminus P$
3: **for** $i \in \overline{S}$ **do**
4:     Pick an arbitrary $j_i \in \mathsf{top}_q(i, P) \setminus S$
5:     $P_S \leftarrow P_S \cup \{i\} \setminus \{j_i\}$
6: **end for**
7: **return** $P_S$

---

We see that every individual in $S$ is included in $P_S$ as in Line 5 the algorithms ensures that each such individual is included and is never excluded afterwards. Hence, as each individual is chosen in $S$ with probability at least $q/n$, we can see that fairness(RANDOMREPLACE$_q$) $\geq q/k$.

Now, we prove that for any $S \in \mathcal{S}_q$ and any individual $i' \in N$,

$$c_q(i', P_S) \leq c_q(i', P) + \max_{i \in S} c_q(i, P) \leq c_q(i', P) + \sum_{i \in S} c_q(i, P). \tag{1}$$

The second inequality holds as the maximum is at most the sum. Therefore, we focus on the first inequality. If $c_q(i', P_S) \leq c_q(i', P)$, then it trivially holds. Otherwise, $c_q(i', P_S) > c_q(i', P)$. In this case, we can show that there exists some $i \in S \setminus P$ such that $d(i', i) \geq c_q(i', P_S)$ and $j_i \in \mathsf{top}_q(i', P)$. First, note if for each $i \in S \setminus P^*$, $j_i$ does not belong in $\mathsf{top}_q(i', P)$, then it is not possible that $c_q(i', P_S) > c_q(i', P)$, since $\mathsf{top}_q(i', P) \subseteq P_S$. Next, suppose for contradiction that for every $i$ that was included in $P_S$ when $j_i \in \mathsf{top}_q(i', P)$ was excluded from it, it holds that $d(i', i) < c_q(i', P)$. Then, in $P_S$ there are $|\mathsf{top}_q(i', P) \setminus \cup_{i \in S \setminus P}(\{j_i\} \cap \mathsf{top}_q(i', P))|$ individuals that have distance at most $c_q(i', P) < c_q(i', P_S)$ from $i'$ and $|\{i \in S \setminus P : j_i \in \mathsf{top}_q(i', P)\}|$ individuals that have distance less than $c_q(i', P_S)$ from $i'$. Note that

$$| \cup_{i \in S \setminus P} (\{j_i\} \cap \mathsf{top}_q(i', P))| = |\{i \in S \setminus P : j_i \in \mathsf{top}_q(i', P)\}|,$$

and hence we get that there are at least $|\mathsf{top}_q(i', P)| = q$ individuals in $P_S$ with distance strictly less than $c_q(i', P_S)$ from $i'$, which is a contradiction.

From the above observation , we have that

$$c_q(i', P_S) \leq d(i', i) \leq d(i', j_i) + d(j_i, i) \leq c_q(i', P) + c_q(i, P) \leq c_q(i', P) + \max_{i \in S} c_q(i, P)$$

where the penultimate inequality holds because $j_i \in \mathsf{top}_q(i', P)$ and $j_i \in \mathsf{top}_q(i, P)$ from the definition of $j_i$. This proves (1).

Summing (1) over all $i' \in N$, we have SC$_q(P_S) \leq$ SC$_q(P) + n \cdot \sum_{i \in S} c_q(i, P)$. Taking the expectation of this equation with respect to $S$, and using the fact that $\Pr[i \in S] = q/n$, we have

$$\mathbb{E}[\mathrm{SC}_q(P_S)] \leq \mathrm{SC}_q(P) + n \cdot \sum_{i \in N} \frac{q}{n} \cdot c_q(i, P) = (q + 1) \cdot \mathrm{SC}_q(P),$$

that is, $\frac{1}{q+1} \cdot \mathbb{E}[\mathrm{SC}_q(P_S)] \leq \mathrm{SC}_q(P)$. Combined with the assumption that $P$ is $\alpha$-representative, i.e., $\alpha \cdot \mathrm{SC}_q(P) \leq \min_{P' \in S_k(N)} \mathrm{SC}_q(P')$, we get repr$_q$(RANDOMREPLACE$_q(P)) \leq \frac{\alpha}{k+1}$. $\qquad\square$

The next theorem shows that when $q = \Omega(k)$, RANDOMREPLACE$_q$ attains an asymptotically optimal representation-fairness tradeoff.

**Theorem 5.** *For any $q \leq k/2$ such that $k \bmod q = 0$, every selection algorithm $\mathcal{A}_{k,q}$ with* fairness$(\mathcal{A}_{k,q}) \geq q/k$ *satisfies* repr$_q(\mathcal{A}_{k,q}) \leq k/q^2$.

*Proof.* Let $m = k/q$. Consider an instance with $n > 2k$ individuals on the real line, where one individual $i_0$ is located at 0, $\lceil n/m \rceil - 1$ people are at 1, and at least $\lfloor n/m \rfloor$ people are located at

each position $j \in \{2, \ldots, m\}$. This way, there are at least $n/m - 1 = (n/k) \cdot q - 1 \geq 2q - 1 \geq q$ individuals located at each $j \in [m]$.

Any optimal panel would include $q$ individuals from each position $j \in [m]$, which results in a $q$-cost of 1 for $i_0$ and a $q$-cost of 0 for the rest. Hence, $\mathrm{SC}_q(P^*) = 1$. However, any selection algorithm with fairness of at least $q/k$ selects $i_0$ with probability of at least $q/n$. When $i_0$ is selected, there must exist a group $j \in [m]$ from which the algorithm selects at most $q - 1$ people, incurring a $q$-cost of 1 for at least $n/m$ people (person $i_0$ and at least $n/m - 1$ people at position $j$). Hence,

$$\mathbb{E}[\mathrm{SC}_q(\mathcal{A}_{k,q})] \geq \frac{q}{n} \cdot \frac{nq}{k} = \frac{q^2}{k},$$

which completes the proof. $\qquad\square$

From the above theorem, it follows that Algorithm 1 achieves the highest possible fairness of $q/k$ subject to positive representation.

## 5    Experiments

Next, to complement the worst-case analysis of the selection algorithms we have considered so far, we conduct an empirical comparison between the average-case $q$-social cost and representation ($\mathrm{repr}_q$) of the different selection methods.

**Datasets.** Data on the metric-structure preferences of groups in their full richness are difficult to come by, but it is reasonable to expect that the extent to which individuals feel well-represented by one another is at least partly a function of their relative characteristics along some observable features. We therefore begin with two datasets that express the characteristics of populations along a range of features, and randomly construct synthetic metric preferences from these feature signatures.

`Adult` *dataset.* Our first source of demographic data is the UCI `Adult` dataset, which was derived from the 1994 Current Population Survey of the US Census Bureau, and is made available by the UCI Machine Learning Repository under a CC BY 4.0 license [27]. It contains a range of demographic variables principally related to employment. Our experiments do not require `Adult` to be representative of any actual population, nor should this an assumption be made lightly [28]. For `Adult` we choose the features `workclass`, `education`, `marital status` and `sex`. `Adult` contains $n = 30162$ individuals with values for each feature, who may be viewed as a distribution over the 721 unique feature vectors which they collectively hold.

`ESS` *dataset.* Our second source of demographic data is the *European Social Survey* (ESS) [29], which is made available by the Norwegian Centre for Research Data under a CC BY 4.0 license. We use the ESS Round 9 (2018) data, which consists of 46,276 people in 27 countries, and contains $\sim$1450 features regarding socioeconomic demographic, political beliefs, geographical region, house-hold composition, personal values, media use and trust, etc. Most of the features are country-specific, which leaves roughly 250 features available per country, while each country has between 781 and 2745 entries (with a mean of 1713). Each entry is assigned an analysis weight which is aimed to correct the differential selection probabilities. In contrast to our experiments with `Adult`, we use all of the available features available in `ESS`. We report the experiment results based on the data of the United Kingdom (2204 entries). Similar results were obtained using the data from other countries.

**Metric construction.** We choose some set of features along which to evaluate the individuals in the population. These are the features which will inform our metric. For each feature $s$, $F^s$ is the set of possible values that this feature can take. Each individual $i$ is then represented as a vector of feature values, where $f_i^s$ is the value that $i$ has for feature $s$. For *categorical* features (e.g., marital status, sex), we define $d(i, j; s) = 1 - \mathbb{1}\{f_i^s = f_j^s\}$. For *range* (e.g., income) features we define $d(i, j; s) = |f_i^s - f_j^s|/(\max_{i',j'} |f_{i'}^s - f_{j'}^s|)$, where the normalization makes $d(i, j; s) \in [0, 1]$. Then, for each feature we sample a weight $w_t \sim U[0, 1]$ uniformly at random from the interval $[0, 1]$. Finally the distance between individuals $i$ and $j$ and our metric $d$ is defined to be $d(i, j) := \sum_s w_s \cdot d(i, j; s)$.

For the empirical evaluation of our selection algorithms on these randomly generated metrics, we suppose that this is in fact the distribution for a population large enough that there are at least $k$ individuals with any given feature vector. As these metrics represent populations, the $q$-social costs in Figures 1 and 2 are normalized by population size. Figures 1 to 3 depict data averaged over 100 random metrics constructed in this manner, and the error bars show the standard error of the mean.

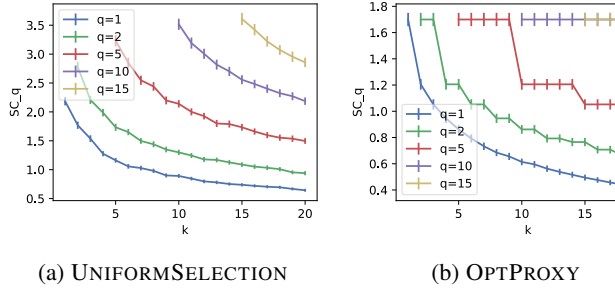

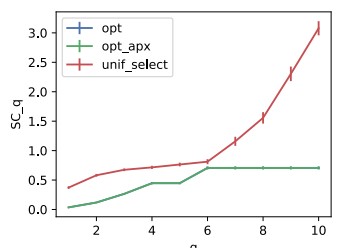

(a) UNIFORMSELECTION  (b) OPTPROXY

Figure 1: The $q$-social cost of UNIFORMSELECTION and OPT-PROXY for $k \in [1, 20]$ and a selection of $q$, based on `Adult`.

Figure 2: OPTPROXY finds the *optimal* panel $SC_q(P^*, d)$ on small instances $k = 10$, based on `Adult`.

**Proxy for the optimum.** In evaluating UNIFORMSELECTION and RANDOMREPLACE, we use a proxy for the optimal $q$-social cost, since $n$ and $k$ are too large to support finding $SC_q(P^*, d)$ exactly. This selection algorithm OPTPROXY (indicated as 'opt_apx' in legends) is an implementation of the fault-tolerant metric $k$-medians algorithm of Kumar et al. [26], which guarantees a constant-factor approximation to $SC_q(P^*, d)$. This algorithm uses a constant-factor metric $k$-medians algorithm as a primitive; we implement the local search algorithm of Arya et al. [30] with single swaps. More specifically, OPTPROXY works as follows. It finds the (approximately) optimal $\lfloor k/q \rfloor$-median solution and picks $q$ individuals from each of these $\lfloor k/q \rfloor$ positions, and then selects the remaining $k \bmod q$ people uniformly at random.

To evaluate the fidelity of OPTPROXY we compare it with $SC_q(P^*, d)$ for 100 metrics $d$ constructed by drawing 30 randomly chosen feature vectors from the supports of `Adult` instances described above. For panels of size $k = 10$ and all values of $q$ we find that OPTPROXY recovers $SC_q(P^*, d)$ exactly (Figure 2). The fact that the $q$-social cost attained by OPTPROXY for fixed $q$ and $q \in (k/2, k]$ appears constant, on the other hand, is not universally true of $SC_q(P^*, d)$. These plateaus are due to the way OPTPROXY selects panels by returning $q$ copies of the optimal 1-median location. Since $P_q^*$ need not be of this form, it is interesting that it exhibits this same step-like behavior in Figure 2.

**Experimental results.** In Figure 1a, we see UNIFORMSELECTION behaving as expected. For a fixed $k$, the $q$-social cost of a uniformly random panel is reliably higher for larger $q$, and for a fixed $q$, it decays smoothly as we increase the panel size $k$. Under the same conditions, OPTPROXY behaves similarly (Figure 1b) but the decay is not as smooth due to the reasons explained above. Here, all of our approximately optimal panels $P_q$ start at the same value of $SC_q(P_q, d)$ when $q = k$ and decay as $q$ remains fixed and the size of the panel $k$ increases.[2]

Figure 3 shows the ratio between the $q$-social cost obtained by different selection algorithms and the $q$-social cost of OPTPROXY, which is an approximation of their representation, averaged over 100 randomly generated metrics. Specifically, we evaluate UNIFORMSELECTION and RANDOMREPLACE$_r$ (indicated as 'unif_select' and 'rand_replace_r' in legends, respectively), the latter being a generalization of RANDOMREPLACE$_q$ that replaces $r$ individuals instead of $q$ in the input panel and achieves representation at least $\alpha/(r + 1)$ (when the input panel has representation $\alpha$) and fairness at least $r/k$. We compare these algorithms for a range of $r$ and $q$, on the `Adult` dataset (Figure 3a) with panels of size $k = 20$ and on the `ESS` (Figure 3b) dataset with panels of size $k = 40$.

*Representation of* UNIFORMSELECTION. As expected, the representation of UNIFORMSELECTION is the lowest for all $k$ and $q$. However, it still achieves consistently good representation across the board, with representation dipping below 50% only on the `Adult` dataset with $q$ close to $k$.

We find it especially interesting that, on both datasets, UNIFORMSELECTION exhibits good representation in the regime of $q \leq k/2$, as this contrasts with our worst-case result. We conclude that, in real-world datasets, UNIFORMSELECTION may be quite representative even with respect to the

---

[2]This first property is to be expected: when $q = k$ and points in the metric may appear on the panel multiple times (as is the case with these distributions), an optimal panel $P_q^*$ consists of $k$ copies of a (carefully chosen) single point in the metric. For these panels $P_q$, OPTPROXY is simply finding that point.

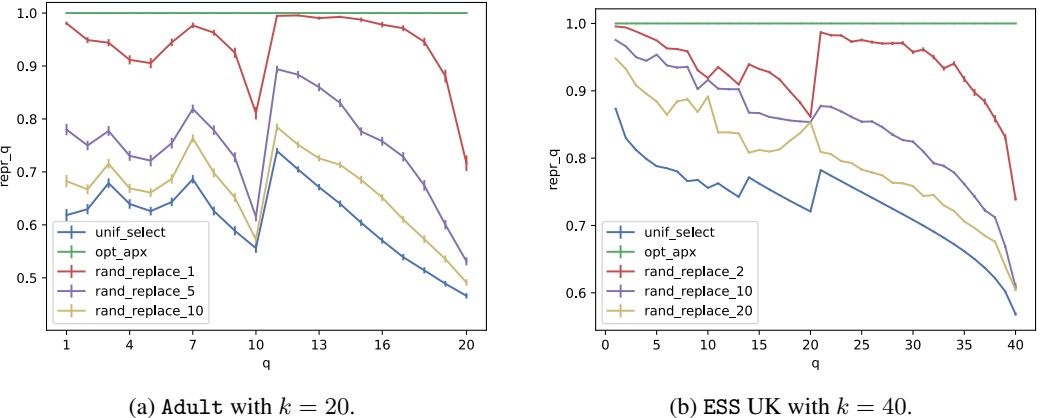

(a) `Adult` with $k = 20$.

(b) `ESS UK` with $k = 40$.

Figure 3: Comparison of different algorithms for fixed $k$, where RANDOMREPLACE$_r$ is applied to the panel selected by OPTPROXY. The $y$-axis shows the average ratio of the $q$-social cost of OPTPROXY to that of different algorithms.

$q$-social cost measure when $q \leq k/2$. Overall, we view these results a providing empirical support for the use of UNIFORMSELECTION in practice.

*Representation of* RANDOMREPLACE. Figure 3 shows that RANDOMREPLACE$_r$ achieves significant improvements in representation compared to UNIFORMSELECTION for small values of $r$ — at the cost of fairness, of course. This tradeoff may be justified for some values of $q$, but the advantage of RANDOMREPLACE is much smaller around $q = k/2$ and when $q$ approaches $k$, so for these regimes RANDOMREPLACE should not be in the running.

*Shift at* $q = k/2$. In both Figures 3a and 3b, there is a sudden improvement in the representation of UNIFORMSELECTION at $q = k/2$. This provides empirical support for the peculiar transition in the $q$-social cost occurring $q = k/2$, which is predicted by the theory and is the reason we separated Sections 3 and 4.

## 6   Discussion

Our results in Section 4 show that in some cases, relaxing fairness requirements allows improving representation dramatically. More generally, it is interesting to understand the tradeoff between representation and fairness, and to chart the Pareto frontier. In Appendix B, we take some first steps in this direction. One observation is the aforementioned generalization of RANDOMREPLACE that replaces $r$ individuals instead of $q$. We also show that a random-dictatorship-like algorithm gives nontrivial fairness and representation guarantees in the regime of $q > k/2$. However, there remain several open questions: for example, when $q \leq k/2$, what level of fairness can be achieved if we seek constant representation?

We focused our attention on the $q$-cost formulation of Caragiannis et al. [13], in which each individual measures their distance to the $q$-th closest panel member. One can analyze the representation-fairness tradeoff with other cost functions. For example, what if different individuals use different values of $q$? Another appealing choice is when each individual measures the *average* distance to all panel members; in Appendix C, we show that for this cost function, uniform selection achieves representation of at least $1/2$.

More broadly, one can use measures of representation other than the *total cost* to all individuals. For example, one may wish to use a selection algorithm that minimizes the *maximum cost* to any individual, or strikes a balance between maximum cost and total cost. We hope that answers to some of these questions will lead to a better understanding of the strengths of sortition, and to new ways of realizing this democratic paradigm.

**Acknowledgements.** This work was partially supported by the National Science Foundation under grants IIS-2147187, CCF-2007080, IIS-2024287, and CCF-1733556; by the Office of Naval Research under grant N00014-20-1-2488; and by an NSERC Discovery Grant.

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
