# OpenReview forum: "Is Sortition Both Representative and Fair?"
_NeurIPS.cc/2022/Conference — NeurIPS 2022 Accept_

### Official Review · Reviewer_MDMp · 2022-06-22

**Rating:** 7
**Confidence:** 4
**Soundness:** 4 excellent
**Presentation:** 4 excellent
**Contribution:** 3 good

**Summary:**

The paper addresses the problem of sortition, which is a democratic process consisting in randomly selecting a subset of citizens from a population. There are two important criteria in this selection process: *fairness*, meaning that citizens should have an equal chance to participate, and *representation*, meaning that the chosen panel should well represent the large population. The authors introduce a representation metric which is inspired by distortion-based analysis of voting over metric spaces, and which depends on a parameter $q$. Depending on the value of $q$, the paper provides results on whether a perfectly fair algorithm achieves good representation. In the regime where there is a necessary fairness-representation trade-off, an algorithm with guarantees on the two criteria is presented.
The results are illustrated with experiments on the Adult dataset, as well as the European social survey.


**Questions:**

* Can you please respond to the concerns raised above?
* Related to the main weakness I described, what are the approximate runtimes of the algorithms? What happens for larger values of $k,N$, e.g. $k=100$ as for citizens assemblies?




**Limitations:**

The social context of this work is well described.

**Strengths And Weaknesses:**

Strengths:
* The paper is clear, well-motivated and well-written.
* The idea of measuring representation based on the distance of an individual to the $q$-closest member is clever and intuitive.
* The theoretical results are not technically difficult, but they are intuitive, simple to understand and definitely interesting.
* The experimental evaluation is convincing enough.
* I appreciated that the authors anticipated the need for a better understanding of the fairness-representation trade-off in the Appendix.

Weaknesses:
* Algorithm 1, Line 1: There is no explanation on how to compute the minimizer over all possible subsets of size $k$ among the set of size $N$. The minimization problem is likely intractable; therefore the authors should describe a computationally efficient method to solve this. I found no details in Section 4, neither in the experimental section, nor the Appendices.

Minor comments:
* Topic: The paper is not related to machine learning: neither the techniques, nor the application problem. However, even if the paper falls outside NeurIPS usual topics, I do think sortition is an important problem.
* Additional references on representation constraints when selecting a panel of representatives from a set of candidates (if you think they are off-topic for your work, please explain why). In particular, [2,3] also use a representation metric in that context.

[1] L. Elisa Celis, Lingxiao Huang, and Nisheeth K. Vishnoi. 2018. Multiwinner voting with fairness constraints. In Proceedings of the 27th International Joint Conference on Artificial Intelligence (IJCAI'18). AAAI Press, 144–151.

[2] Lang, J., & Skowron, P. (2016). Multi-Attribute Proportional Representation. Proceedings of the AAAI Conference on Artificial Intelligence, 30(1). https://doi.org/10.1609/aaai.v30i1.10024

[3] V. Do, J. Atif, J. Lang, N. Usunier. Online Selection of Diverse Committees. IJCAI 2021, Aug 2021, virtual, Canada.

---

> ### Author Response · Authors · 2022-07-29
> **Response to Review of Paper8106 by Reviewer MDMp**
>
> We respond to the reviewer's concerns below:
>
> > Algorithm 1, Line 1: There is no explanation on how to compute the minimizer over all possible subsets of size $k$ among the set of size $n$. The minimization problem is likely intractable; therefore the authors should describe a computationally efficient method to solve this. I found no details in Section 4, neither in the experimental section, nor the Appendices.
>
> This criticism is well taken, and we have been able to effectively address it with a little more work. Please see the green text in Section 4 of the rebuttal pdf.
>
> Finding the optimal panel is indeed hard to approximate in general; metric $k$-medians is the special case when $q=1$. We had chosen to present RandomReplace using the optimal panel to show the limits of existential results, since it is not a priori clear that the representation-fairness tradeoff that it achieves is attainable even given unbounded computation.
>
> The revised Theorem 4 has been generalized to show that, asymptotically, the same tradeoff is attained by using a panel that is a constant-factor approximation to the optimal panel. Crucially, the algorithm OptProxy, which we introduce in Section 5, returns such a panel in polynomial time. In summary, we can get representation $\Theta(1/q)$ and fairness $q/k$ in polynomial time. (We also find that RandomReplace, starting from OptProxy, performs quite well in our experiments.)
>
> > Topic: The paper is not related to machine learning: neither the techniques, nor the application problem. However, even if the paper falls outside NeurIPS usual topics, I do think sortition is an important problem.
>
> We thank the reviewer for their open-mindedness. It is worth mentioning that the paper "[fair sortition made transparent](https://proceedings.neurips.cc/paper/2021/file/d7b431b1a0cc5f032399870ff4710743-Paper.pdf)" (reference [4] in the paper) also isn't related to machine learning, yet it was presented in last year's NeurIPS. The scope of NeurIPS in recent years has been expanding well beyond machine learning and there is great interest in topics like sortition (see also reference [3]).
>
> > Additional references on representation constraints when selecting a panel of representatives from a set of candidates (if you think they are off-topic for your work, please explain why).
>
> These references are relevant, and we thank the reviewer for pointing them out. We now briefly discuss the suggested papers in the revised version of the paper; see the red text in Section 1 of the rebuttal pdf.
>
> > Related to the main weakness I described, what are the approximate runtimes of the algorithms? What happens for larger values of $k, N$, e.g. $k=100$ as for citizens assemblies?
>
> Uniform selection is of course quite efficient, with the exact runtime depending on how one implements uniformly sampling $k$ indices from $[n]$ without replacement.
>
> To answer the reviewer's question for RandomReplace with OptProxy as a starting panel, we ran a new experiment with parameters that are typical for large citizens' assemblies. We used the ESS dataset and a standard laptop (quad core, 1.7GHz CPU, 16GB RAM). For $n\approx 2200$, the computation took less than 15 seconds for $k=100$, which includes computing the distance matrix (5 seconds) and running OptProxy (7 seconds). OptProxy itself scales well for practical values of $k$:
>
> | Value of $k$     | OptProxy runtime |
> | ----------- | ----------- |
> | 20   | <4s      |
> | 100  | <7s        |
> | 200  | <10s        |
> | 500  | <20s        |
>
> We will add these results to the appendix.

---

> > ### Comment · Reviewer_MDMp · 2022-08-08
> > **Thank you for your detailed response**
> >
> > I greatly appreciate the effort that the authors put on their response and the paper update. My main concern was the computation of Line 1 in Algorithm 1, which has been well addressed in the updated version of the paper. The explanation is clear. I also appreciate the additional experiments for larger values of $k$, which are convincing, and the reactions to my other minor concerns. I am raising my score to 7.

---

> > > ### Author Response · Authors · 2022-08-09
> > > **Thank you**
> > >
> > > We greatly appreciate your attention to our rebuttal and revision.

---

### Official Review · Reviewer_6NwB · 2022-07-04

**Rating:** 8
**Confidence:** 3
**Soundness:** 3 good
**Presentation:** 3 good
**Contribution:** 3 good

**Summary:**

The paper considers the fairness and representativeness of sortition, a randomized method of group selection. Typically sortition involves uniformly random selection of k participants. This work asks whether other methods of selection may lead to a more representative panel with minimal impact on fairness. Fairness here is the goal of selecting each participant with at least probability k/n while representation is the ability of selecting a panel such that each non-member is similar to a member. In particular, they make use of the "q-social cost" metric -- the sum of distances from each individual to their q-th closest panel member.

The paper shows that when q > k/2 uniform random selection provides optimal representation. For smaller q the paper shows that good representation is not possible with only a tiny loss in fairness but develops an algorithm that modifies a low social cost panel to improve representation. Experiments are performed showing the efficacy of the algorithms they have developed. For both Uniform Selection and RandomReplace, several examples are given demonstrating various q-social costs for different values of k and q.

**Questions:**

1. Is it the case that RandomReplace is meant to provide a good balance of fairness and representation as compared with the optimal panel? Assuming so, is that evident in the experiments? I am unclear on what the experiments say about RandomReplace other than that it is (quite understandably) not as low-cost as the optimal panel.

2. Section 3 begins by alluding to multiple algorithms that are perfectly fair. Are there multiple such algorithms? Aren't they all necessarily equivalent to Uniform Selection?

**Limitations:**

The authors have provided only very little discussion of the potential societal impact of their work. However, the work relatively far removed from meaningful real-world impact; I do not feel such discussion is missing, only absent.

**Strengths And Weaknesses:**

I find the idea of the q-social cost, once I got some intuition for it, quite a fitting idea for measuring the quality of a panel formed via sortition. Focusing on metrics beyond just fairness is useful and relatively novel in sortition. The paper is generally quite well written although it would be nice to have a little bit more emphasis on giving natural, intuitive explanations of your definitions of representation (primarily) and fairness (fairly clear already). I found the proofs of the theorems in the main paper generally clear and they seem correct.

Due to the simplicity/natural usage of the Uniform Selection algorithm adding to theoretical knowledge on the mechanism is a useful contribution.  The RandomReplace algorithm is interesting although it seems destined to remain theoretical; finding an optimal panel is obviously a very complex task in a real-world setting. Discussing the importance of the optimality of the panel would be a useful contribution -- e.g. I'm unclear on what properties you would expect if you applied RandomReplace to a uniformly selected panel.

I greatly appreciate the experiment section and some visualization of how q-social cost changes with k and q. Some parts of the experiements section could be clarified (see below).


Minor Issues

Line 127: Should it read "denotes the distance between i and j, and..."?
Line 209: Is P* only defined in Algorithm 1? Mentioning it in text here or in the Preliminaries would be ideal.
Line 218: "The algorithm reaches this line when *it* considers ..."
Line 224: This is the only place you say "agent"; consistency in terms is ideal.

Experiment section:
Is there a reason not to standardize the y-axis? It makes comparison between plots much simpler
What is opt_apx?
In general, more "prettified" legend terms would be better.
In Figure 1c is opt directly under opt_apx? This may be what you mean on lines 297-298 but it could be more clear.

---

> ### Author Response · Authors · 2022-07-29
> **Response to Review of Paper8106 by Reviewer 6NwB**
>
> In response to the reviewer's questions:
>
> > Is it the case that RandomReplace is meant to provide a good balance of fairness and representation as compared with the optimal panel? Assuming so, is that evident in the experiments? I am unclear on what the experiments say about RandomReplace other than that it is (quite understandably) not as low-cost as the optimal panel.
>
> Yes, RandomReplace is meant to provide a balance between fairness and representation. Importantly, its fairness ($q/k$) is given by Theorem 4, and while this is presented as a lower bound, it is essentially tight on any instance (not just in the worst case). Therefore, the experiments focus on the representation of RandomReplace.
>
> We agree that the take-away message from the experiments was not clear. In response to Reviewer fAHs, we have revised the presentation of the experiments in the rebuttal pdf (see the blue text in Sections 1 and 5). In particular, we now note the following regarding RandomReplace in Section 5: "*Figure 3 shows that, for small values of $r$, RandomReplace achieves significant improvements in representation compared to UniformSelection — at the cost of fairness, of course. This tradeoff may be justified for some values of $q$, but the advantage of RandomReplace is much smaller around $q=k/2$ and when $q$ approaches $k$, so for these regimes RandomReplace should not be in the running.*"
>
>
> > Section 3 begins by alluding to multiple algorithms that are perfectly fair. Are there multiple such algorithms? Aren't they all necessarily equivalent to Uniform Selection?
>
> Yes, there are multiple such algorithms and no, they are not necessarily equivalent to uniform selection. For example, suppose there are 4 agents, {1,2,3,4}, and we need to select two of them. Selecting the panel {1,2} with probability 1/2 and the panel {3,4} with probability 1/2 is perfectly fair, as each agent is selected with probability 1/2. But it clearly isn't equivalent to uniform selection, which would give probability 1/6 to each of the 6 possible panels.

---

### Official Review · Reviewer_fAHs · 2022-07-05

**Rating:** 5
**Confidence:** 4
**Soundness:** 3 good
**Presentation:** 3 good
**Contribution:** 3 good

**Summary:**

The authors study how representative are panels chosen via sortition, i.e., by selecting randomly individuals from a population. The authors measure representation as follows. First, they assume that there is some metric that measures similarity between members of the population and assume that each person would like to be represented by someone as similar as possible. Then, they say that the q-cost of a panel for a person is the distance of the q-th closest panel member from that person (according to the metric). The cost of a panel is the sum of the q-costs. Finally, representation is the worst-case ratio between the minimum panel cost and the expected one (we look at the expectation because panels are drawn by a randomized algorithm). The authors show that if q is greater than half of the size of the panel then good representation values are possible, and otherwise they are not.

**Questions:**

Why is it important to measure the distance between the candidates, instead of assuming that each agent can give a boolean value indicating if another person does or does not represent him/her?

**Limitations:**

I do not believe this question really applies to the theoretical part of the paper. For the experimental part, it is not clear to me what the actual point is, so I cannot really answer.

**Strengths And Weaknesses:**

Strengths:
- theoretical results are nice and complete

Weaknesses
- The added value of the paper, as compared to previous publication, is nice but not as large as one might expect
- The value of the experiments is very unclear. I could not really tell what the point of the experiment is. The authors crunch some numbers and show plots, but I am missing actual high-level conclusions that go beyond analyzing the shapes of the plots
- Plots in Section 5 are missing any sort of standard deviation/variation information

Minor issues:
l. 99: "99 urban" --> "90 urban" ?
It might be useful to explain in lines 143-151 that correlations in the randomized selection process are allowed (assuming they are).
l. 221: "that that"
l. 222: "individuals" -> "individual"

---

> ### Author Response · Authors · 2022-07-29
> **Response to Review of Paper8106 by Reviewer fAHs**
>
> We believe that we can address the reviewer's main concerns in a satisfying way.
>
> > The value of the experiments is very unclear. I could not really tell what the point of the experiment is. The authors crunch some numbers and show plots, but I am missing actual high-level conclusions that go beyond analyzing the shapes of the plots.
>
> This criticism is very well taken. In response, we have completely revised the presentation of the experimental results in Sections 1 and 5, in order to focus on take-away messages and avoid dry descriptions of the figures. This is reflected in the rebuttal pdf, where the relevant changes are marked in blue. (The results themselves remain unchanged.)
>
> For convenience, we include the new summary of the experimental take-aways (in Section 1 of the rebuttal pdf) here: "*Finally, we evaluate the average-case representation of uniform selection and RandomReplace based on two real demographic datasets. For $q > k/2$, the experiments show that uniform selection achieves representation consistently greater than 55%. Even for $q \leq k/2$, in contrast to our worst-case result, uniform selection achieves good average-case representation. These results suggest that, in practice, uniform selection may be a good choice for all $q$-cost distance functions. While RandomReplace outperforms uniform selection across the board in terms of representation (at the expense of fairness), its advantage is small for values of $q$ around $k/2$ and close to $k$, which means it does not offer a good representation-fairness tradeoff in those regimes. Lastly, for both algorithms, we observe a spike in representation at $q=k/2$, which demonstrates that the chasm between the cases of $q>k/2$ and $q\leq k/2$ is not just a theoretical curiosity.*"
>
> > Plots in Section 5 are missing any sort of standard deviation/variation information
>
> Again, this is well taken. The figures in Section 5 of the rebuttal pdf now all include error bars, which show the standard error of the mean (sem), as mentioned in the orange text.
>
> > Why is it important to measure the distance between the candidates, instead of assuming that each agent can give a boolean value indicating if another person does or does not represent him/her?
>
> The reviewer's suggestion is reasonable; in fact, the literature on approval-based committee elections does assume that agents either approve or disapprove potential candidates. But our framework has important advantages. Indeed, representation is much more nuanced than binary numbers, and evidence shows it depends on intersections of features. For example, all else being equal, an older woman may rank three potential representatives in the order: older woman > younger woman > younger man. Our representation metric takes this one step further and quantifies the intensity of this preference.
>
> From a practical viewpoint, organizers of citizens' assemblies care deeply about representation in terms of multiple features: to achieve descriptive representation (lines 32-35), they explicitly stratify over features like gender, age, ethinicity, education and geography (lines 48-53). Therefore, it is crucial that the representation metric be able to capture these issues.
>
> In addition, note that our main theoretical result, Theorem 1, is oblivious to the representation metric; in particular, there isn't even a need to elicit information about representation from agents, which is another advantage over the approach proposed by the reviewer.

---

> > ### Comment · Reviewer_fAHs · 2022-08-09
> > **Answer**
> >
> > Thank you for the answer. While I remain somewhat unenthusiastic, I certainly appreciate the revision and I am adjusting my scores.

---

> > > ### Author Response · Authors · 2022-08-09
> > > **Thank you**
> > >
> > > We greatly appreciate your attention to our rebuttal and revision.

---

### Meta-Review · Area_Chair_w4iU · 2022-08-26

**Recommendation:** Accept
**Confidence:** Certain

**Metareview:**

Reviewers like the problem of fair sortition for its importance as well as its fitness to NeurIPS. Reviewers also liked the solid theoretical results. Minor concerns were raised about the technical depth and uninformative experiments, but the overall sentiment is quite positive.

**Award:**

No

---

### Decision · Program_Chairs · 2022-09-14

Accept